

# Photos provide information on age, but not kinship, of Andean bear

Russell C. Van Horn[1], Becky Zug[2], Robyn D. Appleton[3,4], Ximena Velez-Liendo[5], Susanna Paisley[6] and Corrin LaCombe[1,*]

[1] Institute for Conservation Research, San Diego Zoo Global, San Diego, CA, USA
[2] Nelson Institute for Environmental Studies, University of Wisconsin–Madison, WI, USA
[3] Department of Forest and Conservation Sciences, University of British Columbia, BC, Canada
[4] Spectacled Bear Conservation Society, Squamish, BC, Canada
[5] Centro de Biodiversidad y Genética, Universidad Mayor de San Simon, Cochabamba, Bolivia
[6] Durrell Institute of Conservation and Ecology, University of Kent, Canterbury, Kent, UK
* Current affiliation: We Thrive Global, Reno, NV, USA

## ABSTRACT

Using photos of captive Andean bears of known age and pedigree, and photos of wild Andean bear cubs <6 months old, we evaluated the degree to which visual information may be used to estimate bears' ages and assess their kinship. We demonstrate that the ages of Andean bear cubs ≤6 months old may be estimated from their size relative to their mothers with an average error of $<0.01 \pm 13.2$ days (SD; $n = 14$), and that ages of adults ≥10 years old may be estimated from the proportion of their nose that is pink with an average error of $<0.01 \pm 3.5$ years ($n = 41$). We also show that similarity among the bears' natural markings, as perceived by humans, is not associated with pedigree kinship among the bears ($R^2 < 0.001$, $N = 1,043$, $p = 0.499$). Thus, researchers may use photos of wild Andean bears to estimate the ages of young cubs and older adults, but not to infer their kinship. Given that camera trap photos are one of the most readily available sources of information on large cryptic mammals, we suggest that similar methods be tested for use in other poorly understood species.

Corresponding author
Russell C. Van Horn,
rvanhorn@sandiegozoo.org

## INTRODUCTION

The Andean bear (*Tremarctos ornatus*, FG Cuvier) is endemic to diverse habitats across a broad latitudinal range in Andean South America but it is vulnerable to extinction (*Goldstein et al., 2008*). Although it is likely that the global population of this bear is declining dramatically due to habitat loss, fragmentation, and poaching (*Goldstein et al., 2008*), we know little of its ecology (*Garshelis, 2004*), demography (*Garshelis, 2011*), and genetic structuring (*Viteri & Waits, 2009*), making it difficult to plan for its conservation. To facilitate research in support of Andean bear conservation we've assessed whether we can estimate the ages and assess the kinship of individual Andean bears. Because conservation success may be improved through engagement of local people (*Byers, 1999*; *Danielsen et al., 2007*; *Peyton, 1989*), and because local people may have knowledge and

skills beneficial to scientific research (*Sharma, Jhala & Sawarkar, 2005*; *Stander et al., 1997*; *Zuercher, Gipson & Stewart, 2003*), we've focused on methods that rely on a minimum of technology.

Individual appearance may provide information not only on identity (e.g., *Van Horn et al., 2014*) but also on age and even kinship, in species as disparate as giraffe (*Giraffa camelopardalis*; *Berry & Bercovitch, 2012*; *Foster, 1966*) and lions (*Panthera leo*; *Whitman et al., 2004*). Age in other bears has been inferred, with some error, by morphological measurements and dental cementum annuli (*Bridges, Olfenbuttel & Vaughan, 2002*; *Christensen-Dalsgaard et al., 2010*; *Costello et al., 2004*; *Marks & Erickson, 1966*; *McLaughlin et al., 1990*; *Medill et al., 2009*; *Mundy & Fuller, 1964*; *Stoneberg & Jonkel, 1966*; *Willey, 1974*), but noninvasive methods of age estimation have not been developed for bears. It appears that the markings of some young Andean bears may become less prominent during maturation and that many Andean bears grizzle during aging, but such changes are not obviously consistent or predictable enough to allow age estimation (*Van Horn et al., 2014*). In addition, because monitoring the changes in markings would require repeated assessments across years, monitoring those changes to estimate age would not be feasible for short-term or cross-sectional demographic research. Because the point-in-time estimated size of offspring relative to their mothers may predict their age (e.g., *Jongejan, Arcese & Sinclair, 1991*), we evaluated whether such data predicted the ages of young Andean bear cubs. In addition, because point-in-time samples of nose color are a reliable but potentially sexually-dimorphic indicator of age in another carnivore (*Panthera leo*, *Whitman et al., 2004*), we examined the degree to which the nose color of Andean bears reflected their age. Genetic analysis would provide strong evidence of kinship (e.g., *Woods et al., 1999*) and genetic tools are being developed for Andean bears (e.g., *Viteri & Waits, 2009*), but collection of genetic samples is not always feasible in the humid tropical forests and grasslands where most Andean bears are thought to live (*Goldstein et al., 2008*). Aside from genetic data, kinship may be inferred from similarity of appearance among individuals in some species in some studies (*Pan troglodytes*, *Gorilla gorilla*, *Mandrillus sphinx*, and *Papio ursinus*, *Alvergne et al., 2009*; *Cygnus columbianus*, *Bateson, Lotwick & Scott, 1980*; *Acinonyx jubatus*, *Caro & Durant, 1991*; *Macropus giganteus*, *M. rufogriseus*, *Jarman et al., 1989*; *P. troglodytes*, *Parr & De Waal, 1999*; *Vokey et al., 2004*), but not in others (*A. jubatus*, *Kelly, 2001*). The inheritance of markings among bears is poorly understood (*Higashide, Miura & Miguchi, 2012*) and there is some evidence that patterns in markings of Andean bears are not obviously heritable (*Eck, 1969*), so the link between kinship and similarity in markings among Andean bears is uncertain, at best. We therefore assessed whether this link is informative. If information on an Andean bear's age and kinship can be extracted from its appearance, then non-invasive methods such as camera traps may provide elusive information that is valuable for conservation.

## MATERIALS AND METHODS

We extracted information from portraits of captive Andean bears of known identity, age, and pedigree that were posted online, and from zoo personnel and field researchers in

North America, Europe, and South America (*Van Horn et al., 2014*). If we did not know the date on which the photograph was taken, we assigned it the midpoint of the time period in which the photo was taken (e.g., photos taken in 'July' were assigned the date 15 July).

## Visual estimation of age through relative body size

To evaluate whether the relative size of young cubs might predict their age, we extracted information from opportunistically-collected photographs of known-age cubs born in captivity, and from young cubs found in their natal dens in the tropical dry forest of northwest Peru (6°26′S, 79°33′W), where research on Andean bear ecology and behavior has been underway since 2007. We located active natal dens by inferring den entry from the sudden cessation of new telemetry positions and by then searching near the last previous transmitted locations, along with searching similar sites during the same season. We estimated the ages of cubs found in their natal dens from their development (e.g., ability to lift head, eyes closed or open, ability to stand, ability to walk), when compared to published descriptions of captive cub development (*Aquilina, 1981*; *Bloxam, 1977*; *Malzacher & Hall, 1998*; *Molloy, 1989*; *Müller, 1988*; *Peel, Price & Karsten, 1979*; *Saporiti, 1949*; *Stancer, 1990*). We later opportunistically collected photos of some of these same wild cubs and their mothers with camera traps set during a long-term study. Within those camera trap photos we measured the size of cubs, relative to the size of their mothers, for wild-born and captive-born cubs that were <180 days old. We chose this criterion as a conservative estimate of the age period within which the growth of male and female bears appears similar and approximately linear (*Bartareau et al., 2012*; *Blanchard, 1987*; *Bridges, Olfenbuttel & Vaughan, 2002*; *Kingsley, 1979*; *McRoberts, Brooks & Rogers, 1998*) and because growth among older cubs might be influenced by factors other than age (e.g., seasonal or interannual variation in food availability). To avoid potentially confounding variation that might be introduced by variation in litter size we also excluded data from twin litters. We estimated the relative sizes of cubs by taking the mean of three replicate measures of the same fixed post-cranial measurement of cubs and their mothers when they were the same distance from the camera, as determined by visual landmarks in the photographs ($n = 14$, $2.0 \pm 1.3$ photos/mother-cub pair). Each of the three replicate measures was itself the mean of three measurements by each of three observers. To reduce the impact of measurement error we only estimated the relative size of limb segments that we thought would be most visibly discrete (i.e., clearly defined by joint or bone protuberance) and repeatable between cubs and their mothers. We excluded photographs in which matching measurements could not be made on both a cub and its mother; due to the opportunistic nature of the photographs those measurements differed among mother-cub pairings: lower hindleg ($n = 6$), lower foreleg ($n = 4$), shoulder height ($n = 3$), and upper hindleg ($n = 1$). We constructed candidate predictive models of relative cub size from cub age (69–180 days), cub provenance (captive-born or wild-born), cub identity (4 captive-born, 3 wild-born), and the interaction between cub identity and age. We then used an information theoretic approach (*Burnham & Anderson, 2002*) to compare

the candidate models using $AIC_c$ as the key criterion for model selection, and we used $R^2$ and $p$ to assess the effectiveness of the 'best' model for describing a cub's relative size.

## Visual estimation of age through nose color

To investigate the relationship between the color of a bear's nose and its age (years) we first screened photos of captive Andean bears to exclude photos that did not show the entire nose, photos that did not appear in focus when magnified to 2–4X, and photos from which there were <1,000 pixels in the image of the nose. To avoid non-independence between photos we excluded multiple photos of the same bear that were taken within 365 days, and we renamed the 76 remaining photos from 58 bears (32M, 26F), aged 0.3–31.4 years, with random numbers. We then expressed the color of a bear's nose as the proportion of the area of the nose that was pink ('proportion pink') by taking the mean of three independent replicate estimates by the same observer (i.e., the lead author) of the proportional area of pink in each photo, excluding the nostrils (which were often shaded), and excluding pink scar tissue. We had longitudinal series of photos from 10 bears (7M, 3F) that provided 12 pairwise within-individual comparisons of the change in nose color over time; the average annual change in the proportion pink was $0.02 \pm 0.02$, which lent credence to the use of proportion pink as an indicator of age. Because there were multiple photographs for some but not all bears ($1.31 \pm 0.6$ photos/bear), to avoid non-independence of data and to allow for model testing we randomly selected 1 photograph per bear from photographs of bears ≥9.9 years old (the minimum age at which we observed pink on the nose) for use in model building and retained the other data in this age range for use in model testing. We then used the proportion pink as the response variable in linear regression analyses with candidate models including age, sex, and the interaction of age and sex. We used $AIC_c$ as the key criterion for model selection, with $R^2$ and $p$ to assess the effectiveness of the 'best' model for describing the proportion of the nose that was pink. When multiple candidate models were competitive (i.e., $\Delta AIC_c \leq 2$), we used full model averaging (e.g., *Lukacs, Burnham & Anderson, 2009*) to derive the predictive equation including age, sex, and the interaction of age and sex. To assess the fit and putative power of relationships predicting age we then examined the reverse relationships, with age as the response variable, and examined the distribution of the relevant residuals. When possible we tested the ability of equations to predict the ages of bears in images that had not been used to describe the relationship between age and nose color.

## Similarity of markings and kinship

To assess humans' ability to visually evaluate kinship among Andean bears, as part of a larger study, we created an online survey and sought volunteer participation by colleagues, peers, personal contacts, and a solicitation in the International Bear News (*Paisley et al., 2010*; *Van Horn et al., 2014*). We asked participants to rate the similarity of 11 pairs of images of bears whose kinship was unknown to them; the average pedigree $r$-values across these pairs of images was $0.32 \pm 0.23$. Participants were asked to rate the similarity of the markings of bears in these images as 1 of 5 categories: exactly the same, similar, slightly different, extremely different, and unable to determine. Participants ($n = 109$) in the online

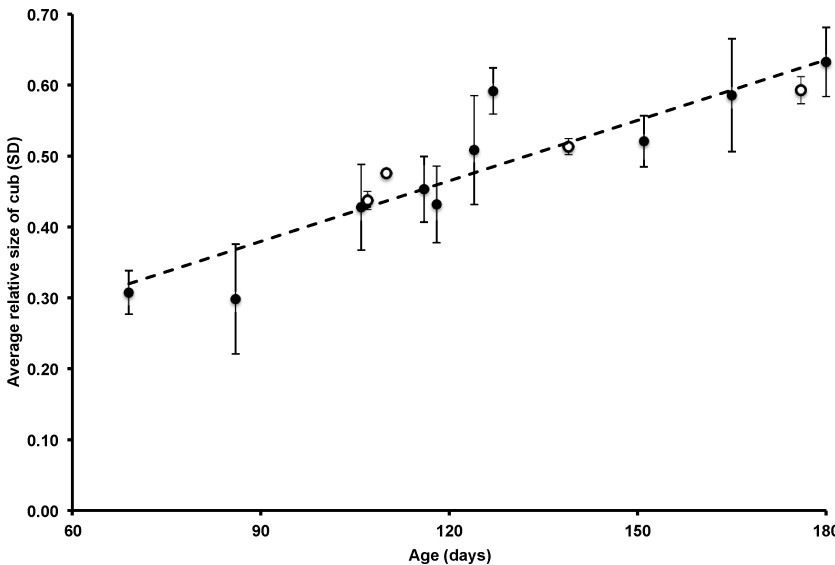

**Figure 1 The size of Andean bear cubs in proportion to the size of their mothers while the cubs were 2–6 months old.** The dashed line illustrates the linear regression of average relative size in response to age in days among 4 captive-born (open circles) and 3 wild-born (filled circles) cubs.

survey rated the similarity of, on average, $9.6 \pm 1.7$ of 11 pairs of images. We used ordinal logistic regression to examine the strength of the relationship between the perceived visual similarity of markings and the pedigree $r$-values of the bears in the images with candidate models including pedigree $r$-values, whether the participant had experience working with Andean bears ($n = 10$) or not ($n = 99$), and the interaction of pedigree $r$-value and experience. We used $AIC_c$ as the key criterion for model selection, with $R^2$ and $p$ to assess the effectiveness of the 'best' model for describing the relationship between perceived visual similarity and the pedigree $r$-values.

Unless otherwise noted all quantities are expressed as $\bar{x} \pm SD$, and statistical significance refers to two-tailed $p = 0.05$. Statistical analyses were conducted in JMP 10.0.2 (SAS Institute Inc., Cary, NC). Animal research was approved by the IACUC committee of San Diego Zoo Global (#10–023).

## RESULTS

### Visual estimation of age through relative body size

The model that described cub relative size from only an intercept and cub age in days ($R^2 = 0.835$, DF $= 13$, $p < 0.001$) fit the data better than all other models that included combinations of cub identity, cub provenance, and an interaction term (i.e., $\Delta AIC_c > 4$): relative size $= 0.123 + 0.003 * (\text{age in days})$. This model would not perform well for much younger and older cubs, as it predicts that newborn cubs are 12.3% of their mother's size and that cubs would be the same size as their mothers when only 313 days old, but from 2–6 months in age there appears to be a linear relationship between cub age and relative size (Fig. 1). The reverse relationship is (age in days) $= -15.263 + 293.26 * (\text{relative size})$, from which the average residual was $1.32 \times 10^{-14} \pm 13.2$ days.

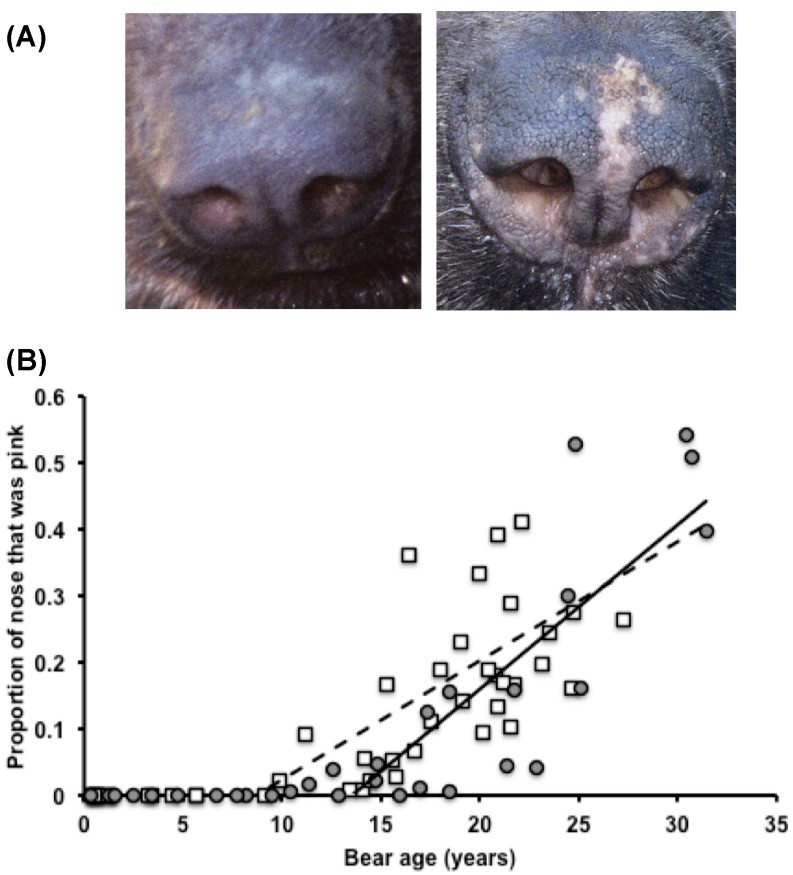

**Figure 2 The proportion pink of an Andean bear's nose across age in (A) a male Andean bear ('Tommy', studbook #264), aged 2 years and 17 years, and (B) in 76 photos of 58 captive-born Andean bears (32M, 26F).** The trendlines show the relationships between the proportion pink and age in males (open square, dashed line) and females (filled circle, solid line). Photo credit: San Diego Zoo Global.

## Visual estimation of age through nose color

No pink was seen on the nose in 26 photos of 20 bears (10M, 10F), of which all but one (96.1%) were <10 years old; the youngest age at which we saw pink on the nose was 9.9 years ($n = 50$ photos, $16.54 \pm 14.7$% pink; Fig. 2). Nearly all of the 52 photos of bears >9.5 years old (96.2%) showed some pink on the nose. There was variation among the repeated estimates of the proportion pink from those photographs (i.e., average SD of proportion pink across repeated estimates = 2.1). The linear model, built upon data from 41 photos of 41 bears (23M, 18F), which best fit the data predicted the proportion pink from only age ($R^2 = 0.554$, DF = 39, $p < 0.001$) but there were two other competitive models (i.e., $\Delta$AICc > 2): the model that also included sex, and the model that included sex and the interaction of sex and age. We therefore used model averaging to derive the equation (proportion pink) $= -0.257 + 0.022*$ (age in years) $+ 0.0006 *$ (age in years) $* (z)$ where $z = 0$ if male or $z = 1$ if female. However, in practice it will not always be possible to determine the sex of a bear from camera trap photos. The best predictive model for bears of unknown sex predicted the (proportion pink) $= -0.254 + 0.022*$ (age in years). In reverse,
this relationship predicted (age in years) $= 15.055 + 25.129*$ (proportion pink) with an average residual of $2.99 \times 10^{-15} \pm 3.46$ years. Testing this model with the 7 independent data points (6M, 1F) yielded an average error of $-1.62 \pm 2.3$ years. Using the 23 points from males in the model-building data set, we found that for males (proportion pink) $= -0.156 + 0.018*$ (age in years) ($R^2 = 0.335$, DF $= 22$, $p = 0.004$). The reverse of this relationship predicted age (in years) of males as $15.482 + 10.698*$ (proportion pink) with an average residual of $2.39 \times 10^{-15} \pm 3.2$ years. Testing this model with the 6 independent data points from males produced an average error of $-2.43 \pm 2.7$ years. Using the 18 points from females in the model-building data set, we found that among females the (proportion pink) $= -0.33 + 0.0245*$ (age in years) ($R^2 = 0.703$, DF $= 17$, $p < 0.001$). The reverse relationship predicted for females that (age in years) $= 15.435 + 28.644*$ (proportion pink) with an average residual of $-1.28 \times 10^{-15} \pm 3.7$ years. With only 1 independent data point from a female we cannot further assess the errors in age estimation that result from this model.

### Similarity of markings and kinship

Markings of Andean bears vary greatly even among full siblings (e.g.,  Fig. 3). The average pedigree $r$-values across pairs of test images was between the kinship levels of half-siblings and full-siblings, yet the average similarity rating given to these paired images by participants was $3.38 \pm 0.85$, between 'slightly different' (i.e., '3') and 'extremely different' (i.e., '4'). There was not a meaningful relationship between the pedigree $r$-values of bears and similarity rankings of their photos across all participants ($R^2 < 0.001$, $N = 1,043$, $p = 0.499$), among the participants who had worked with Andean bears ($R^2 < 0.001$, $N = 98$, $p = 0.843$), or among the participants who had not worked with Andean bears ($R^2 < 0.001$, $N = 945$, $p = 0.436$). In addition, the model that best described the relationship between pedigree $r$-values of bears and similarity rankings of their photos included only an intercept term. Models including either pedigree $r$ or experience working with Andean bears were also both competitive (i.e., $\Delta$AICc $< 2$) but none of these three models fit the data well (i.e., each had $R^2 < 0.001$). Thus, there is no evidence that similarity among paired images, as perceived by experienced or inexperienced participants, reflected pedigree kinship among the bears.

### DISCUSSION

By reviewing photos of cubs <6 months old we found that for the first several months after young Andean bears leave their natal dens, the relative size of cubs can be used to predict their age and then estimate their birthdates. At present the only data on birthdates of Andean bears come from captivity (e.g., *Spady, Lindberg & Durrant, 2007*) and from 2 dens in Ecuador (*Castellanos, 2010*; *Castellanos, 2015*), so the estimation of any additional birthdates of wild cubs should offer important insights into Andean bear reproductive ecology. Interestingly, because the provenance of young cubs had no impact on their relative growth, relative growth of young cubs should be stable across habitats, allowing the use of this relationship to predict ages and estimate birthdates across the species' range. Given that a similar method of age estimation is effective in a phylogenetically distant

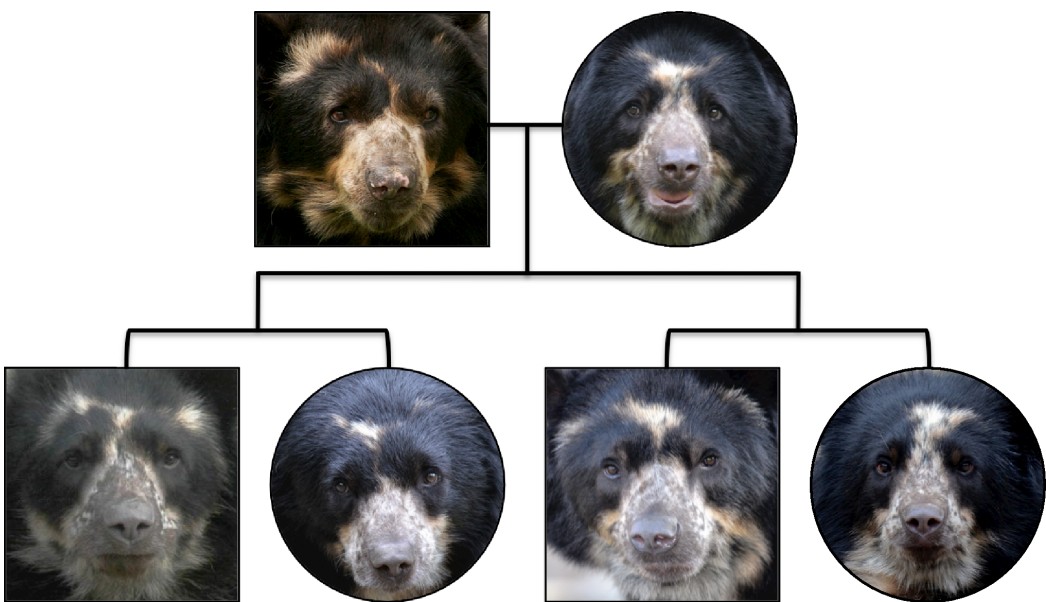

**Figure 3 A photographic pedigree of captive-born Andean bears.** Squares represent males and circles represent females in this pedigree of male 'Nikki' (studbook #415, 19.3 years old), his mate 'Billie Jean' (studbook #748, 7.4 years old), and their four offspring: the littermates 'Bernardo' (studbook #837, 1.2 years old) and 'Chaska' (studbook #838, 3.3 years old), and the littermates 'Curt' (studbook #860, 2.0 years old) and 'Nicole' (studbook #861, 1.3 years old).

species (*Ourebia ourebi*, *Jongejan, Arcese & Sinclair, 1991*), we think the relative size of dependent offspring may be a useful way for investigators to visually estimate the ages and birthdates of progeny in many other species.

Nose color provides a clear noninvasive indicator of whether an Andean bear is older or younger than 10 years: if any of the nose is pink the bear is almost certainly >10 years old, and vice versa. We do not know whether Andean bears begin undergoing other physiological, behavioral, or ecological changes at this age, but older Andean bears also show grizzling on their faces (*Van Horn et al., 2014*). Using the proportion pink of the bear's nose, and whatever information is available about a bear's sex, we can estimate the age of a wild Andean bear to ± 3–4 years. Because we did not have independent samples for model building and testing, we cannot predict well the precision of age estimates generated with other sample sets, but these estimates are less precise than age estimates for some other bear species (e.g., *Christensen-Dalsgaard et al., 2010*; *Costello et al., 2004*). However, those estimates require capture and handling of the bear, while measuring nose color does not. It may not be easy to obtain many suitable photos of the noses of free-ranging bears without the use of lures and relatively complex configurations of cameras traps, but two of us (RVH, RDA) have done so. We do not know if nose color changes in a predictable manner in other bears and in other carnivores except lions (*Whitman et al., 2004*)), although we have seen photos of some cats (e.g., *Leopardus pardalis*, *Puma concolor*) showing variation in their nose color. We therefore suggest that nose color may provide valuable information on age structure in other carnivores.

Our data indicate that it is not possible to infer kinship among Andean bears based on the perceived similarity of their markings. This is consistent with *Eck*'s (*1969*) hypothesis that patterns in markings are not heritable and this affirms that genetic tools (e.g., *Viteri & Waits, 2009*) are needed to infer kinship among wild Andean bears.

Although the methods we describe cannot replace long-term research on known individuals, we believe that they will facilitate the collection of data and enhance the value of camera trapping efforts for the conservation of Andean bears. Because these methods require relatively little advanced technology or training, we hope that they will permit the engagement of local people in this research. In addition, we believe the examination of the relationships among relative size, nose color, sex, and age among known-age individuals of other species may produce similarly useful methods across more taxa.

## ACKNOWLEDGEMENTS

We thank the Smithsonian's National Zoological Park and the following for the use of their photographs in Fig. 3: Daniel Reidel ('Billie Jean', 'Chaska', and 'Nicole'), Tracey Barnes ('Bernardo' and 'Curt'), and Valerie Abbott ('Nikki'). We thank Drs. Peter Arcese and Ron Swaisgood for their intellectual contributions. We thank the volunteer participants in the online survey, which was made possible by Yuri Nataniel Daza (Universidad de San Simon). The following zoological gardens and individuals shared photographs of captive Andean bears for use in the online survey: Antwerp Zoo (Sander Hofman), Basel Zoo (Dr. Friederike von Houwald), Brookfield Zoo, Cheyenne Mountain Zoo, Cincinnati Zoo, Cleveland Metroparks Zoo, Connecticut's Beardsley Zoo, Dortmund Zoo (Dr. Florian Sicks), Durrell Wildlife Conservation Trust (Mark Brayshaw), Gladys Porter Zoo, Houston Zoo, Köln Zoo (Dr. Lydia Kolter), Minnesota Zoo, Oglebay's Good Zoo, Racine Zoo, Reid Park Zoo, Rolling Hills Wildlife Adventure, Salisbury Zoo, San Antonio Zoo, San Diego Global, Smithsonian National Zoo, and Smoky Mountain Zoo. We were able to collect data on wild Andean bears only through the efforts of Javier Vallejos, José Vallejos, and Isaí Sanchez, assisted by Álvaro García-Olaechea. We thank the citizens of the Rio La Leche watershed for permission and support to work on communal land.

### Funding

RVH and CLC were supported by San Diego Zoo Global, BZ was supported by the US Department of Education, and RDA was supported by the International Association for Bear Research & Management, Wildlife Media, Calgary Zoo, and the Spectacled Bear Conservation Society. The funders had no role in study design, data collection and analysis, decision to publish, or preparation of the manuscript.

### Grant Disclosures

The following grant information was disclosed by the authors:
San Diego Zoo Global.
US Department of Education.

International Association for Bear Research & Management.
Wildlife Media.
Calgary Zoo.
The Spectacled Bear Conservation Society.

## Competing Interests

Russell C. Van Horn is an employee of the San Diego Zoo Institute for Conservation Research. Robyn D. Appleton is an employee of the Spectacled Bear Conservation Society. Corrin LaCombe was an employee of the San Diego Zoo Institute for Conservation Research when the research described in the manuscript was performed.

## Author Contributions

- Russell C. Van Horn and Becky Zug conceived and designed the experiments, performed the experiments, analyzed the data, contributed reagents/materials/analysis tools, wrote the paper, prepared figures and/or tables, reviewed drafts of the paper.
- Robyn D. Appleton, Ximena Velez-Liendo and Susanna Paisley conceived and designed the experiments, performed the experiments, contributed reagents/materials/analysis tools, wrote the paper, reviewed drafts of the paper.
- Corrin LaCombe conceived and designed the experiments, wrote the paper, reviewed drafts of the paper.

## Human Ethics

The following information was supplied relating to ethical approvals (i.e., approving body and any reference numbers):

The Research Compliance Office of Miami University has reviewed the human subjects portion of this research (as project 01632e) and ruled it exempt from full IRB review, under Exempt Category 2.

## Animal Ethics

The following information was supplied relating to ethical approvals (i.e., approving body and any reference numbers):

Animal research was approved by the Institutional Animal Care and Use Committee Zoological Society of San Diego and governmental approval was granted by the Dirección General Forestal y de Fauna Silvestre (Peru). Approvals were granted as IACUC #10-023 and Resolución Directoral No. 0245-2012-AG-DGFFS-DGEFFS.

## Data Deposition

The following information was supplied regarding the deposition of related data:

Dryad: DOI 10.5061/dryad.77q9r.

## Supplemental Information

Supplemental information for this article can be found online at http://dx.doi.org/10.7717/peerj.1042#supplemental-information.

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
