# Peer review of "Photos provide information on age, but not kinship, of Andean bear"

_PeerJ, doi:10.7717/peerj.1042_

## Round 0.1 · original submission · Minor Revisions

· Academic Editor

Minor Revisions

I was fortunate to receive four expert reviews of your MS. All of the reviewers had favorable impressions of your MS but request further clarification on several points that they articulate quite clearly. Reviewer 3 also questions the accuracy of some of your citations. Please correct for accuracy rather than dropping the references in question.

I agree with Reviewer 2 that it would be ideal to have data comparing the accuracy of experts and non-experts. Such an addition would strengthen the contribution of your MS but is not essential for publication. I leave that to your discretion. You might at least provide more detail about the participants who completed your survey - how many were colleagues versus inexperienced observers?

A more general title might capture the attention of a broader group of scientists. For example, something like "Use of visual information to predict age and kinship in Andean bears" or "New Methods for Observational Estimation of Sex and Kinship in Andean Bears". I also leave this to your discretion.

·

Basic reporting

The basic reporting in this article is written clearly and concisely, and has enough basic information to be relevant to all readers interested in this species or other bears, and with a working knowledge of English (this would be understandable for readers in range countries). The introduction and background information is also complete so that it fits for South American generalist wildlife biologists and bear biologists with international experience.

Experimental design

This article does not fail to meet PeerJ standards. It is original primary research, with clearly defined research questions and methods. It provides sufficient information about these methods, and statistical analyses, to be reproducible by others. It is clear from IACUC and IRB review comments that the research was conducted with current ethical standards in our field.

Validity of the findings

I believe the data are robust enough to support the conclusions and discussion in this article. The conclusions are appropriately stated, connected to the original question, and suggests an advance that can be used in non-invasive studies of this species.

Additional comments

I think this is a very creative approach to developing field techniques that can be used to age Andean bears. They are simple and low-tech enough to be used by local field assistants, and the creativity extended to your thinking about non-invasive sampling. Your statistics are very good, and lead to conclusions that are also applicable by field assistants without statistical knowledge but with knowledge about the bears. Congrats.

·

Basic reporting

The article is clear and well-written. The structure is also appropriate for PeerJ. The figures are all relevant and clearly described.

I do have several brief recommendations to improve the basic reporting. The authors mention that not much is known about the Andean bear, but it would be useful for the reader to have a little more background information provided. A brief paragraph at the beginning of the introduction with very basic demographic information would better set the stage for the following content on conservation. For example, the discussion of conservation refers to education of "local" people, but an unfamiliar reader would not know where the local refers to.

Also, in several places, the authors refer to other studies with a "carnivore" or "species." Please be more specific about the species involved in the previous work (e.g., line 40, line 48, line 186).

Experimental design

Overall, the authors have clearly defined the question and their approach. The method section has been described in sufficient detail, but I do have a few clarification questions that would improve the manuscript.
line78 - were the measurement estimates taken via the photographs ever validated with actual in-person measurements (for the captive individuals)? This would be an important validation that the photograph technique accurately assesses size.
line 96 - who performed the three independent estimates?
-An additional analysis to assess experts versus non-experts in identifying similarity between photographs might be interesting. The pattern might hold - providing further support for their conclusion - or it might reveal interesting differences between those who work with Andean bears as compared to other species.

Validity of the findings

The conclusions are logical extensions of their findings. The data appear to have been analyzed appropriately and reveal clear outcomes that are clearly related to the research question.

Additional comments

The authors have presented a clear analysis of several noninvasive approaches to estimating age and relatedness in the Andean bear. These results are straightforward and may be useful for improving conservation-related efforts and studies of individuals in the wild.

·

Basic reporting

Submission is clear and unambiguous. The introduction includes sufficient background (the need for techniques to improve conservation research on this species, the usefulness of low-tech, non-invasive methods such as photography as shown with other species, including age/size and age/coloration and kinship studies), and clearly details how it fits into the field of broader knowledge (example of using photographs for use in studies of wild animals).
There is one error, Line 107-108 the authors state " to assess the effectiveness of the ‘best’ model for describing a cub’s relative size." As this is in the section on nose color, I assume this is an error and should read about percent pink nose color.
Four citations are inappropriate; none seriously detract from the submission overall, but should be changed before acceptance.
Line 27, citation (Van Horn et al. 2014), does not evaluate the use of facial markings in Andean bears to predict age, and indeed reports that "grizzling" appears to be an age predictor. The authors need to correct this reference, either to a study which actually evaluates facial markings and age in Andean bears, or to explain why they believe that it is unlikely that facial markings could be used to predict age, or remove that point from the introduction. However, in light of its relevance to examining nose coloration (and not facial markings), a discussion of this issue in the introduction is warranted. This reviewer would like to know, if facial markings have not been previously evaluated as an age predictor, why this was not included in the present study.
Line 47, Mills et al. 2000 is a study of genetic analysis, not kinship identification from appearance. This citation should be removed.
Line 75, Bridges, Vaughan, & Fox 2011, does not refer to bear cub growth or size. This citation should be removed.
Lines, 73-76: this statement "the growth of bear cubs within the first 6 months of life often appears linear" is not well-supported by the references. Only Bridges et al. 2002 explicitly examines growth of bear cubs with linearity between ages 0-3 months. The rest study bear growth and age more broadly. Bartareau et al. 2012 and Kingsley 1979 show linearity but between ages 1-5 without resolution to infer specifically that 0-6 months is also linear. The 6 month cut-off is not justified by these references; either different references need to be used that represent this statement or an alternative explanation for the age range used should be offered.
Line 197 misses a citation that refutes their statement; the authors state "We do not know if nose color changes in a predictable manner in other bears and in other carnivores". Whitman et al. 2004 shows just this in lions, which the authors themselves note in their introduction. The authors should change this statement to acknowledge Whitman et al. 2004's contribution to changes in nose color predicting age of carnivores.
The submission adheres to templates and structure generally. No conclusions section is used; this reviewer thinks a short summary of the discussion could be useful, but is not necessary. The only departure is that the submission uses two cover pages, both of which list the authors. The first page has title, authors without affiliations, and abstract, while the second contains the author list (again) with affiliations.
Figures are relevant and sufficiently labeled.
Submission is self-contained and an appropriate unit of publication; the authors used photographic appearance of bears to assess multiple possible relationships, such as age from cub relative size, age from nose color, and kinship from facial appearance. This techniques are all of use to for ecological and conservation studies and are appropriately included in the same article.

Experimental design

The submission fits within the Aims and Scope of the Journal.
The submission defines the research question fairly clearly: whether appearance of Andean bears can be used to extract information relevant to conservation studies (age and kinship). This is stated clearly in the abstract, but presented piece by piece in the introduction along with justification for each component. A summary sentence at the end of the introduction would be helpful for making this clearer. This research question is extremely relevant and meaningful with large potential impacts for conservation research of Andean bears and other species.
The investigation was sufficiently rigorous and had a high technical standard; the authors used predictive modeling to assess relationships between age and size and age and nose color. They used logistic regression to evaluate the ability of human raters to assess similarity in facial markings of Andean bears. The study with cub size has a small sample size, but the results are still of interest. The study with nose color has a commendably large sample size of 58 bears. The authors were rigorous in selection of photos for the nose color analysis; photo selection for the cub size and age was not detailed. In addition, the inclusion of both wild and captive bear cubs in the study, considering its potential impact on cub size also demonstrates rigor. The care taken in the study on nose color to exclude dependent data points is also good, although it is not clear if this care was taken for the first study on cub age and size.
The use of length of lower hind limb and shoulder height as measures of size was not explained; the authors should include references or discuss why they thought these measurements were sufficient or most relevant. One study the authors cite that evaluated bear cub growth and age used ear and hair length; while it is understandable that these measurements may not be feasible for photographic evaluation, some justification for the choice of size measurements should be included.

The methods are not sufficiently detailed in all areas for reproducibility. This is particularly so for the first study "Visual estimation of age through relative body size". The authors should include the methods for determining age of the cubs in dens specifically; what measurements were taken that determined age? The authors should also include the method for taking photos of cubs from dens; because size from photos was estimated relative to mothers, it would be important to know how photos of cubs and mothers in dens were collected. In addition, screening processes for photos in this study would be relevant; clearly only photos including both mother and cub are required; the authors do mention that they used photos in which mother and cub were the same distance from the camera using visual landmarks in the photos, but more detail is needed. The authors state that they took 3 repeated measures of the same morphological measurements from photographs; if these were taken by different raters or not is not clear, and a reliability estimate of these ratings would be helpful.
For the second study "Visual estimation of age through nose color", the methods are much better detailed. As in the previous study, reliability estimates and explanation (same rater, different rater) of the repeated estimates of nose color should be included. Exactly what "being alert for the presence of pink scar tissue" means is not clear. Was pink scar tissue not counted as "pink" or were these photos excluded or those pixels coded as a third color?
Methods for the third study "Similarity of markings and kinship" are effectively described.

Validity of the findings

The data is generally robust and controlled. For the first study "Visual estimation of age through relative body size", it appears as if independence of data points was not controlled for, possibly due to the low sample size. If data points are not independent, it should be mentioned and justified, otherwise an explanation as to how dependence between points was controlled for needs to be included. In addition, in the discussion the authors report that cub provenance had no impact on relative size (lines 182-185). The authors report including this in their model in methods, but do not mention it in the results; if this is to be included in the discussion then the appropriate statistics should be reported in the results as well. As well in the discussion, the first sentence should clarify that this is based on the authors results, starting with a "we found that" type of intro, and the limitations of this results need to be stated (i.e. limited to single cubs between the ages of 2-6 months). The authors conclusions about cub size and age predictions are otherwise appropriate.
For the second study "Visual estimation of age through nose color", the data is robust, statistically sound, and controlled; all data on which conclusions are based are presented. The same bears (although using different photos of those bears taken at least 1 year apart) are used for both model creation and testing, the authors should note that this may affect their error estimates. Although the effects are not as strong as in the first study, it is still statistically significant and allows for rough estimation of bear age. The conclusions about this study are appropriately stated, the exception being the incorrect statement that I mentioned in the Basic Reporting section of the review.
For the third study "Similarity of markings and kinship" the data is robust, statistically sound, and controlled and conclusions are limited to data presented. The conclusions are appropriately stated as well.

Reviewer 4 ·

Basic reporting

The introduction could benefit from more direct statements on why the research was conducted and why having this type of information is important. The authors do address this more fully in the discussion however. More specifically stated hypotheses or research questions would help with this.The authors also include the importance of engaging local people in conservation work and that an effort was made to minimize the use of technology in order to achieve this goal, but no further mention is made in the manuscript about this and how it was incorporated into the work being presented. An additional figure that would be useful would more precisely demonstrate the measures taken on cub photographs.

Experimental design

The authors only mention two examples of the measures taken to determine cub age from the photographs, having the full list of measures would allow replication. The way in which the authors report the methods makes it unclear how many cub photographs were from the same cubs and how this could potentially impact results given the small number of photographs used. It would also be useful to include the total number of photographs selected for the nose color model building. The authors included sex as a variable in the modelling, what was the rationale behind this decision? There also appears to be an error on line 107 as the sentence reads "for describing the cub's relative size" but this is in the nose color on adult bear section of the methods.

Validity of the findings

the way in which the findings are reported in the results section, namely the visual estimation of age through nose color, is a bit hard to follow. The age at which the authors report noting pink coloration on the nose is at ~10 years of age, however, as written, it appears that each prediction starts with a number of years around 15. Further clarification or a different way to present the results would be very helpful. The authors state in the discussion that nose color does provide a clear noninvasive way to determine if a bear is younger or older than 10 years of age. Is this a relevant distinction and in what way?

---

## Round 0.2 · accepted · Accept

· Academic Editor

Accept

It appears that you have been responsive to the reviewers’ suggestions. It is fortunate that reviewer 3 reviewed your references so carefully so that the revised manuscript is a more accurate reflection of the literature.

One of the most interesting findings to me is that similarity of markings does not predict kinship. Although you address this finding in a two-sentence paragraph toward the end of the discussion, I wonder if more could be stressed here?
I have one remaining point of clarification. You add a statement to ensure the approval of “animal research” but the real participants in your studies were the humans who rated the photographs. Did you also have approval of the IRB?
Minor grammatical Issues:

Please replace the “which” in “The model which…” with “that” in the first sentence under “Visual estimate of age…” on pg. 8. Likewise, on pg. 10, halfway through the last results paragraph.